# The Effectiveness of Suffruticosol B in Treating Lung Cancer by the Laser Trapping Technique

Mulugeta S. Goangul [1,*], Rance M. Solomon [2], Daniel L. Devito [2], Charles A. Brown [2], James Coopper [2], Daniel B. Erenso [2], Ying Gao [3], Aline Pellizzaro [3], Jennifer M. Revalee [3] and Horace T. Crogman [4,*]

[1]   Physics Department, Addis Ababa University, Addis Ababa 1000, Ethiopia
[2]   Departments of Physics and Astronomy, Middle Tennessee State University, Murfreesboro, TN 37132, USA
[3]   Department of Biology, Middle Tennessee State University, Murfreesboro, TN 37132, USA
[4]   Department of Physics, California State University Dominguez Hills, Carson, CA 90747, USA
\*   Correspondence: mulugeta.setie@aau.edu.et (M.S.G.); hcrogman@csudh.edu (H.T.C.);
      Tel.: +1-310-243-2092 (H.T.C.)

**Abstract:** We used laser trapping to study the effects of suffruticosol B on lung cancer cells. Physical and mechanical changes were found to be statistically significant, with a 63.97% increase over untreated cells and a 79.57% increase over untreated cells after treatment for 3 or 6 h, respectively. The treatment affected the internal structure of the cells, with changes in their elastic properties. The cellular responses showed that treatment with suffruticosol B resulted in the decreased proliferation and invasion of cancer cells. These results suggest that the treatment may be useful in preventing or treating lung cancer.

**Keywords:** suffruticosol B; oligostilbene; optical trapping; lung cancer; radiation





## 1. Introduction

The existence of cancer is characterized by an uncontrollable growth of abnormal cells that disregards the normal rules of cell division. The fate of normal cells is constantly determined by signals indicating whether they should divide, differentiate into another cell or die. These signals lead cancer cells to become autonomous, causing them to grow and multiply uncontrollably, resulting in a tumor [1]. There were 10 million cancer-related deaths worldwide in 2020, the leading cause of death [1,2]. In the United States, the number of cancer deaths per 100,000 persons is 161.4, and lung cancer accounts for most of these [3]. It accounts for approximately 20% of all cancer-related deaths worldwide (1.6 million deaths annually), and an estimated 1.8 million new cases are predicted each year in both developed and developing countries [4].

Studies have shown that radiotherapy combined with chemotherapy significantly improves local tumor control and patient survival rates [3–8]. Despite the fact that chemoradiotherapy is superior to radiotherapy for treating locoregional diseases and improving survival, its efficiency is less than maximal, with only an approximately 15% to 20% 5 year survival rate [9]. The effectiveness of standard chemotherapeutic agents, such as cisplatinum and taxanes, has been improved by their use in combination with one another, selected for their own antitumor activity. As a consequence of this toxicity associated with conventional chemoradiotherapy using these agents, the amount of safe RT can often be reduced. Thus, new strategies that increase antitumor activity and, at the same time, reduce toxicity to normal tissues during chemoradiotherapy are needed. Thus, the search for antitumor drugs that increase the radiosensitivity of tumor cells is on its way.

One approach is the use of the laser trapping (LT) technique, which was invented by Ashkin in the early 1980s and is considered one of the greatest discoveries of the twentieth century [10,11]. In this advanced technique, microscopic particles are captured, translated, and manipulated by radiation force [12–17]. With LT, we can conveniently trap

and manipulate dielectric particles as small as a few nanometers in diameter and as large as a few micrometers in size [12,13]. In addition to reducing radiation-induced tissue toxicity, LT might be able to advance the sterilization of cancerous cells at the same time [7,8]. This technique provides unique means to control the dynamics of small particles and manipulate biological samples at the micron level [17,18]. In the physical and biological sciences, this experiment has played a fundamental role. Our group has demonstrated that LT can be used to determine both ionization energy and charge at the single cell level for BT20 and 4T1 breast cancers, neuroblastomas, and RBCs [7,8,16–20].

In recent studies, researchers have discovered that certain traditional Chinese medicines (TCMs) used for treating cancer carry antitumor agents that increase tissue sensitivity to radio waves and protect normal tissues from radiation damage due to the fact of radiation [4–8]. This new approach is combined with TCMs for single cell ionization and is extended to multiple in 4T1 breast carcinoma cells [9]. The cell line was treated with a naturally occurring compound, 2-dodecyl-6-methoxycyclohexa-2,5-diene-1,4-dione (DMDD), which extracted from the root of *Averrhoa carambola* L. produces a significant reduction in the threshold radiation dose. Multiple cell ionizations were observed and related to the chain effect of ionization by the radiation field [8]. It is well known that the botanical plant (*Paeonia suffruticosa*) has antitumor, antioxidative, and other health benefits. It has been shown in previous studies that *Paeonia suffruticosa*, a naturally occurring plant lignin, inhibits the growth of liver, esophageal and lung cancer cells [21,22]. In this study, we used a suffruticosol B, a naturally occurring oligostilbene isolated from peony seeds [21,22]. It is an antitumor agent that is part of the highly regarded 3,5,4′-trihydroxystilbene family of molecules [23–27].

Radiotherapy relies on the fact that healthy cells much more easily undergo DNA repair than cancer cells. The cell cycle controls genome integrity to prevent genetic changes from being passed to subsequent generations [21,28]. Several studies have demonstrated that malignancies are associated with the deregulation of cell cycle checkpoints, including those in the G1/S and G2/M phases [29]. Thus, cell cycle arrest provides an opportunity for DNA repair to occur, hence inhibiting the replication of the damaged template [30]. Although, a recent study showed that the active fractions in suffruticosol B are cytotoxic towards various cancer cell lines, such as MCF-7, MDA-MB-231, HeLa, A459, and CaOV3, since in an induced G2/M phase cell cycle, the cells are more radiosensitive [21,22].

In previous studies on this compound, the antioxidant and anticarcinogenic activities were associated with it. In a laser trap, cancer cells can be exposed to electromagnetic radiation pressure that may operate as a radiosensitizer so that these bioactive compounds can act as radiosensitizers. We present a laser trapping study in this paper comparing the effects of suffruticosol B treatment on human lung carcinoma (A549) to the untreated group.

## 2. Methods

### 2.1. Plant Material

*P. suffruticosa* seeds were collected in Tongling, Anhui, China, and identified in September 2012. The Chinese Academy of Medical Sciences and Peking Union Medical College have deposited a voucher specimen (2012001) in their Seed Resource Bank. The extraction and isolation of suffruticosol B from the dried seeds of *P. suffruticosa* was described previously [24]. Their structures were characterized by ultraviolet (UV), infrared (IR), and mass and nuclear magnetic resonance (NMR) spectroscopy, and the purities of all compounds were determined to be >95%(Tennessee, USA). The compounds were resuspended in dimethyl sulfoxide (DMSO) (Sigma-Aldrich, St. Louis, MO, USA) at a concentration of 10 mM and stored at 4 °C.

### 2.2. Culturing and Treatment of Cells

The human lung carcinoma cell line A549 was used in this study, which was purchased from the American Type Culture Collection (ATCC, Manassas, VA, USA). A medium of RPMI-1640 (Sigma-Aldrich) containing 10% FBS and 100 U/mL penicillin and streptomycin

was used for the cell culture. The cells were incubated at 37 °C in a humidified atmosphere with 5% $CO_2$, harvested by trypsinization, and diluted in medium to $4.5 \times 10^4$ cells per mL. To each well of a 96 tissue culture-treated plate (Corning Costar), 100 μL of diluted cells was added. The cells were incubated overnight and treated with suffruticosol B for 3 and 6 h. The samples were prepared from the untreated, 3 h treated and 6 h treated cells and placed on a depression slide to measure the cells' response to the laser trapping (LT) force.

### 2.3. Laser Trap Design

A schematic of the experiment setup is illustrated in Figure 1. An infrared laser (ƛ = 1064 nm) is shown paired with an inverted microscope with a high numerical aperture (NA) and a digital camera controlled by computer. The power of the linearly polarized infrared diode laser (LS) was governed by a polarizer combination (p) and a half-wave plate (w). A pair of lenses (L1, L2) with beam expander (BE) were combined to align and expand the laser beam. At the trap port of the inverted microscope (Olympus IX 71), four optical mirrors directed the beam to the dichroic mirror (DM). A 45° angle dichroic mirror caused the reflected light to be perpendicular to the incident light. The beam was reflected into the back of the objective lens (OL) with a good numerical aperture and magnification. Lenses L3 and L4 were positioned so that the trap was on the microscope's focal plane. The microscope was attached to the computer-based piezo-driven stage (PS). Using this stage, the collimated sample was located and manipulated for trapping, while the halogen lamp (HL) provided live 2D images for contrast analysis.

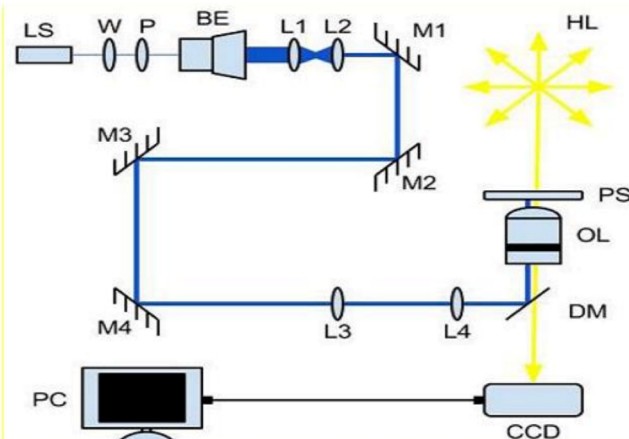

**Figure 1.** A schematic of the experimental setup of the laser trap. The laser's wavelength was 1064 nm and had a maximum power of 8 watts. The power was controlled by the combination of a half-wave plate (W) and a polarizer (P). A beam expander (BE) along with mirrors (M1, M1, M3, and M4) and lenses (L1, L2, L3, and L4) worked together to align and steer the beam as it passed the dichroic mirror (DM) into the back of the objective lens (OL) to manipulate the cell on the piezo-driven stage (PS). A halogen lamp (HL) illuminated the sample so that consecutive images could be taken by CCD camera and analyzed by computer (PC).

### 2.4. Experimental Procedure

From the three prepared samples (2.2), the untreated cancer cell was placed on a dispersion slide and mounted on the PS of the microscope. Three consecutive images of cells were with the digital camera when free, lying on the bottom of the slide. In many cases, the cells adhered swiftly to the slide's coverslip and had to be tapped to separate them for trapping. When the laser port on the microscope was opened, the gate opened and the cell became trapped. A laser gate at the laser port was shut before taking three consecutive images of the cell before releasing it from the trap. As soon as the cell was released, we took images of it every second for a minute. Similarly, the procedure was repeated for the 3 and 6 h treated samples. As part of our research, we compared the effects of laser traps of various strengths on cells that are kept inside the trap, released, and relaxed.

To alter the pressure of the trap, the laser power was varied from 300 until 2700 mW by a 600 mW variation at the laser port right outside of the microscope. Only 30% of this power was applied to the trapped cell, which was outside the microscope's objective lens. We took measurements of ten different cells at each power setting to improve the averages. Images of the cells from the three samples are presented in Figure 2.

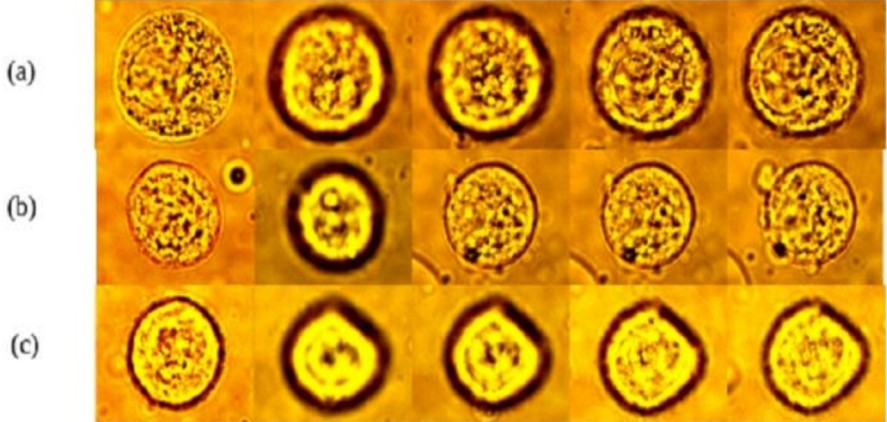

**Figure 2.** (**a**) A549 cell treated for 3 h at 2700 mW; (**b**) A549 cell treated for 6 h at 2700 mW; (**c**) untreated cell at 2700 mW.

The images in the first row (a) show a cell after 3 h of treatment with suffruticosol B; the second row (b) shows a 6 h treated cell; and the third row (c) shows an untreated lung cancer cell. The first two images, from left to right, are the free cell and the trapped cell; however, the last three are sequential images of the relaxation at approximately 3 s intervals.

### 3. Results

#### 3.1. The Physical Properties of the Free Cell

The physical properties of the untreated and treated free cells were examined by statistical analysis of the mean area. The mean area of these cell groups (untreated and treated) was determined from the image captured for each free cell using Image-Pro Plus 6.2, an image processing, enhancement, and analysis software. The area of the cell was measured in pixels square and converted into micrometers square using a conversion factor of 5.28529 and reported in Table 1. The response of the untreated and treated cells with suffruticosol B was investigated. The distribution of the cell size according to four essential statistical parameters (mean, standard deviation, minimum, median, and maximum) can be found in Table 1. As a result of the treatment with suffruticosol B, the average area of the A549 lung cells increased.

The results are displayed in Figure 3a using a histogram. Figure 3a displays the mean area of the cells that were not treated with suffruticosol B (red). These cells had an average area of approximately $186.2 \pm 47.2$ µm$^2$, whereas for the cells treated with suffruticosol B for three hours (blue) it was $304.8 \pm 79.6$ µm$^2$, which was greater than the untreated cells. The change in the average area due to the treatment was observed for the cells treated for six hours (magenta), with an average area of $333.9 \pm 88.2$ µm$^2$.

**Table 1.** The values of the essential statistical parameters of the area of the untreated and treated A549 lung cells free and trapped at 300 mW.

| Treatment Periods (hours) | # of Cells | Mean Area ($\mu m^2$) | Standard Deviation ($\mu m^2$) | Min. Area ($\mu m^2$) | Media Area ($\mu m^2$) | Max. Area ($\mu m^2$) |
|---|---|---|---|---|---|---|
| **Free Cell** | | | | | | |
| 0 | 50 | 186.1 | 47.2 | 92.2 | 186.4 | 311.5 |
| 3 | 50 | 304.8 | 79.6 | 141.9 | 294.5 | 482.6 |
| 6 | 50 | 333.9 | 88.2 | 168.3 | 325.3 | 627.4 |
| **Trapped Cell** | | | | | | |
| 0 | 50 | 134.2 | 36.3 | 68.3 | 124.0 | 133.0 |
| 3 | 50 | 208.8 | 66.7 | 76.9 | 294.5 | 339.8 |
| 6 | 50 | 242.2 | 104.6 | 83.8 | 223.2 | 644.6 |

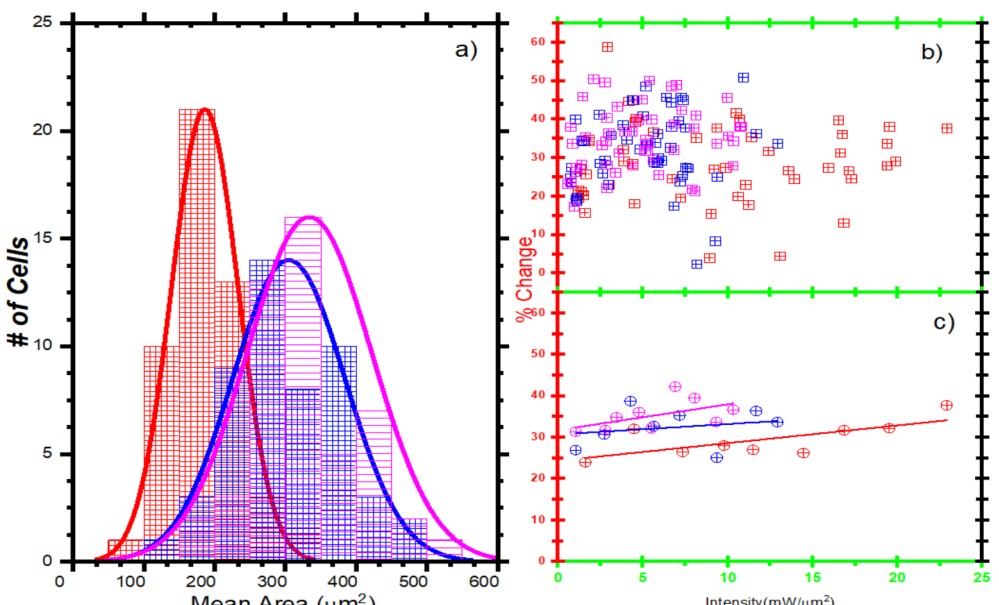

**Figure 3.** (**a**) The size distribution of the mean area of the untreated (red), 3 h treated (blue) and 6 h treated (magenta) cells; (**b**) the percent change of the free cells as a function of the intensity; (**c**) the reduced percent change of the cells as a function of the intensity for a free cell. In all (**a–c**), the traces of the behaviors were made using fourth-order polynomial approximations.

Based on the images from both the free ($A_f$) and trapped ($A_t$) states of the cells, we measured the mean area of each cell. From the measurements we made for the ten different cells, we calculated the average values for $A_t$ and $A_f$. In order to find the relative change in the mean area of the cell at a given power, we used %A = $((A_f - A_t)/A_f) \times 100$. Similarly, we calculated the amount of intensity falling on the trapped cells using the power per mean area of the cell. In Figure 3b, the raw data for the percentage change in the area of cancer cells that were untreated, 3 h treated and 6 h treated are displayed, while in Figure 3c, the relative percent changes in the area are displayed. The average cell area increased with treatment (Figure 3c). In either case, this may indicate that the spherical cells were flattened and became larger or that mitochondrial division occurred after the mitotic phase, resulting in fewer daughter cells.

### 3.2. The Physical Properties of the A549 Cell with Laser Trap

A comparative analysis of the direct laser trapping effects on the A549 untreated and treated cells was also carried out. We first analyzed the relative changes in the trapped cells concerning the corresponding free cells. Following the release from the trap, the relaxation

rates of these same cells were then examined and studied based on their area as well (see Table 2). The size analyses of the images captured for the trapped cells were analyzed with the same procedure as the free cells (Figure 3c).

**Table 2.** The values of the essential statistical parameters of the area of the untreated and treated A549 lung cells with a laser trap at 300 mW.

| | Trapped Cell Mean Area ($\mu m^2$) | | |
| --- | --- | --- | --- |
| | **Untreated** | **Treated** | |
| **Power (mW)** | **0** | **3 h** | **6 h** |
| 300 | 183.3 | 253.5 | 280.9 |
| 900 | 207.9 | 319.1 | 347.2 |
| 1500 | 215.8 | 339.3 | 357.5 |
| 2100 | 234.7 | 340.9 | 344.5 |
| 2700 | 261.9 | 398.3 | 422.6 |
| | Free Cell Mean Area ($\mu m^2$) | | |
| 300 | 186.1 | 304.8 | 333.9 |
| 900 | 225.9 | 357.1 | 364.4 |
| 1500 | 245.1 | 359.7 | 374.4 |
| 2100 | 281.9 | 361.7 | 387.7 |
| 2700 | 297.8 | 399.1 | 440.6 |

The results are displayed in Figure 4a using a histogram. Figure 4a displays the mean area of the cells that were not treated with suffruticosol B. These cells had an average area of approximately (red) $186.1 \pm 47.2$. The area of cells treated with suffruticosol B for 3 h (blue) was $304.8 \pm 79.6$, which was greater than the untreated. A change in the average area due to the treatment can also be observed for the cells treated for 6 h (magenta), with an average area of $333.9 \pm 88.2$.

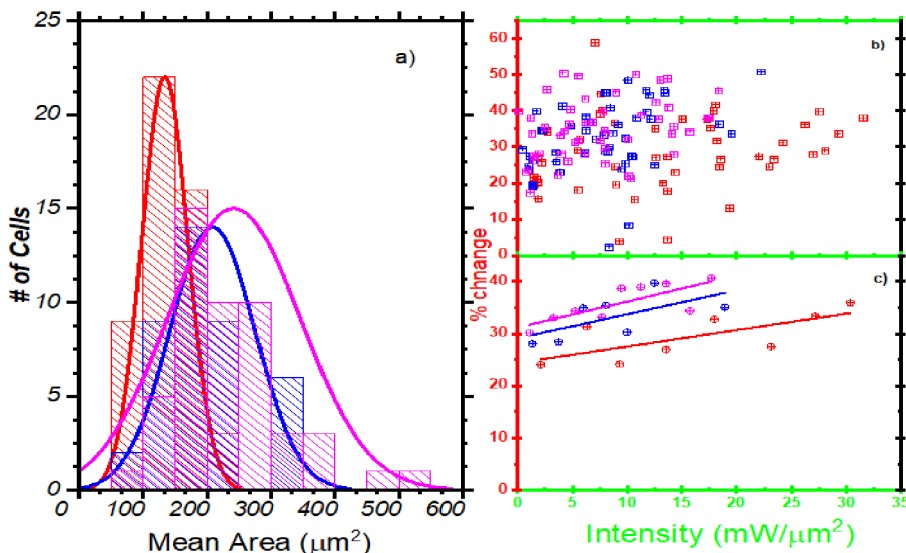

**Figure 4.** (**a**) Size distribution of the mean area of untreated (red), 3 h treated (blue) and 6 h treated (magenta); (**b**) percent change in the trap cells as a function of the intensity; (**c**) reduced percent change in the cells as a function of the intensity for a trap cell. In all (**a**–**c**), the traces of the behaviors were made using fourth-order polynomial approximations.

For the free cells regression, we obtained the following: contribution of the treatment to cell size: B = 22.140, t = 8.176, $p < 0.001$; contribution of the power strength to the cell size: B = 0.39, t = 5.041, $p < 0.001$. For the trapped cells regression, we obtained the following: contribution of the treatment to the cell size: B = 21.637, t = 6.777, $p < 0.001$; contribution of

the power strength to the cell size: B = 0.043, t = 4.678, *p* < 0.001. The results reveal that both treatment length and power strength had a significant impact on cell size, however, more so when the cells were free.

Figure 5 clearly illustrates that in comparing the relative sizes of the untreated and the treated cells, the change in the percent of the mean area of the cells was significantly greater with higher treatment periods. The results in Figures 3 and 4 show that the length of the treatment affected how A549 cells responded to the laser intensity. As the treatment time increased, the relative changes in the cells tended to increase for the laser treatments with intensity. To change the trap pressure, the laser power was varied from 300 to 2700 mW using a 600 mW variation at the laser port right outside the microscope during our laser trapping experiment. Only 30% of this power was directed at the trapped cell, which was located outside the objective lens of the microscope. We measured the area of ten different cells at each power setting, including the laser power that fell on the surface of the cell's area, and calculated the intensities for each trapped cell.

### Cell size as a function of treatment length and power

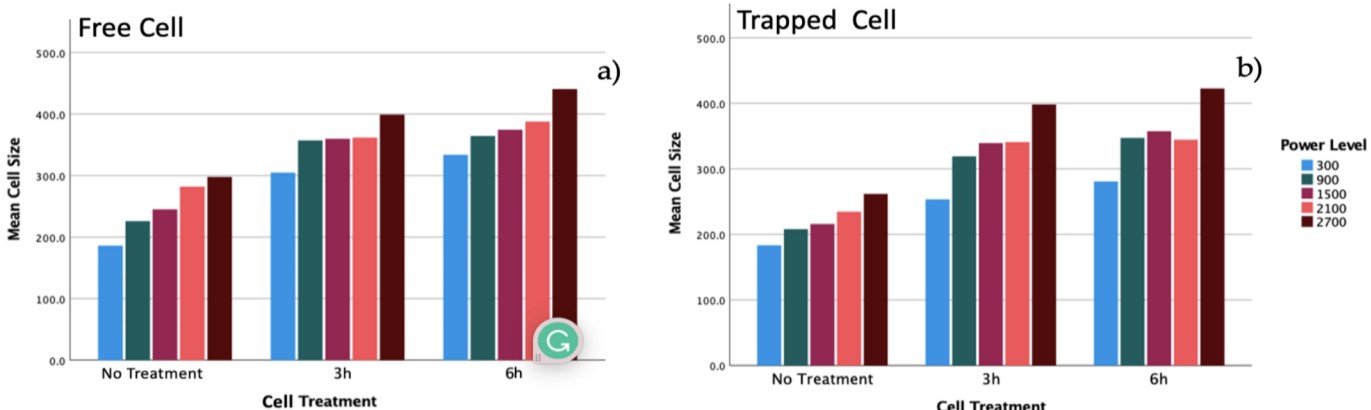

**Figure 5.** Cell size as a function of the treatment length and power: (**a**) mean size of the free cell at different treatment times; (**b**) mean size of the trapped cell at different treatment times.

### 4. Discussion

The observation for the intensity range studied shows that for untreated A549 cells, the change in the cells became larger as the intensity of the trap increased. This is consistent with the generally expected elastic properties of cancer cells compared to free cells. However, the results show that the relative change from untreated to 2 h treated was greater than the relative change from 2 h treated to 6 h treated at the same power. This indicates that suffruticosol B affected significantly the A549 cells, since there were alive, damaged, and dead cells due to the treatment.

When the laser port was opened and power was applied to the treated cancer cells during the laser trapping experiment, the dead cells were easily ejected within a fraction of a second. The slide with untreated cells showed very little cell death compared to the treated slides. This suggests that the drug we used to treat the cells caused cell death. In a recent study, researchers demonstrated that suffruticosol B inhibits the proliferation of cell lines MCF-7 and MDA-MB-231 in human breast cancer through the induction of $G_2/M$ arrest and apoptosis [21]. DNA replication was inhibited by cell cycle arrest, providing a chance for DNA repair.

The time rates of an average area for the untreated and treated cells of five different powers of the trap for suffruticosol B are shown in Figure 6a–c. These three graphs were analyzed in terms of the mean area as a function of the time. Figure 6a represents the time rate at which the untreated cell was released from each power (300–2700 mW), while Figure 6b,c show the release of treated cells from these powers. In Figure 6, red signifies a

power of 300 mW, blue represents 900 mW, green represents 1500 mW, magenta represents 2100 mW, and black represents 2700 mW for each treated and untreated cell. We calculated the total time interval of each cell, using $\Delta t = t_f - t_i$. $t_f$ and $t_i$ are the time taken for the final and the initial relaxing cells; the time interval per frame was found using $R = C\left(\frac{t_f - t_i}{N}\right)$, where C = 0,1,2, 3 . . . (each frame number), and N is the total number of frames.

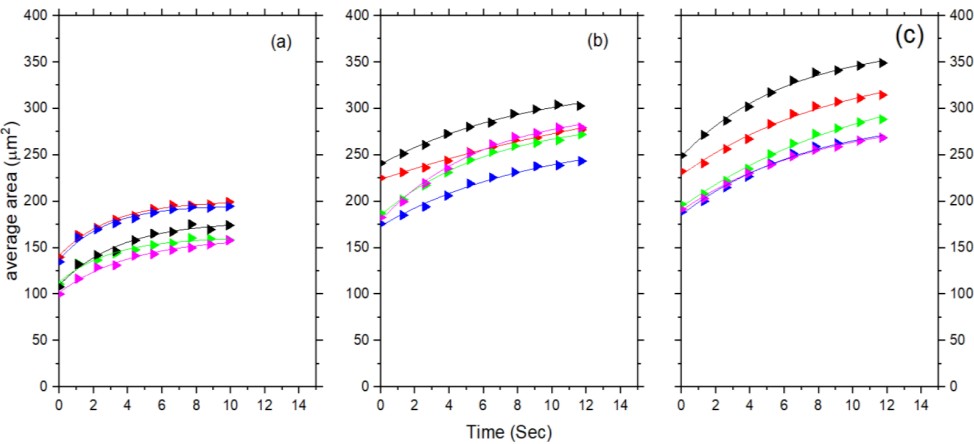

**Figure 6.** Size distribution of the mean area of the cells as a function of the time for untreated (**a**), 3 h treated (**b**) and 6 h treated (**c**) at 300 (red), 900 (blue), 1500 (green), 2100 (magenta) and 2700 mW (black) power.

However, this time rate of the mean area ($A_t$) is the measurement of the mean area of the relaxing cells in one second intervals immediately after the laser power is cut off. For $A_f$, we used the average mean area for the free cell, as previously discussed. Each data point represents the cell released from every five powers calculated using the average of the mean areas (for $A_t$ and $A_f$) for ten different cells. Figure 6 shows that the cell size increased due to the treatment with suffruticosol B. Additionally, the longer treatment of suffruticosol B showed a great spread in size over time (Figure 6b,c). Figure 6 illustrates that the size of the cells was a function of the treatment. There was a clear difference between the treated and untreated cells. For the untreated cells, the power applied mattered less to the cell size. Conversely, as soon as they received treatment, the cells size increased with the increase in power, with a fair difference in the length of treatment. The relationship between the power and cell size, however, seemed more curvilinear than linear as, for example, both the highest (black) and lowest (red) powers seemed to render the highest cell sizes.

Using the laser power as a function of the relaxation rates of the cells in the sample, we defined and calculated the time constant for these behaviors. According to Figure 6, a function was used to calculate the time constant based on the points for the relative change in the area of the maximum. The Origin 2019b programing software was used to fit the data points in Figure 6. An exponential function, $f(t) = A_0 + A_1 e^{kt}$, was founded as shown by the solid lines. The time constant ($\tau = \frac{1}{k}$) is defined as the time in which the relative change in the maximum area immediately after the cell was released from the trap (as predicted by a function f (t = 0)) was reduced by one-third. We then determined a time constant by solving a function for each power.

The average of the three as a function of the trap power is plotted in Figure 7. The data points and the fitting curve shown in red are for the untreated inner lung carcinoma cell, and those shown in blue and black are for the 3 and 6 h treated human lung carcinoma cells, respectively.

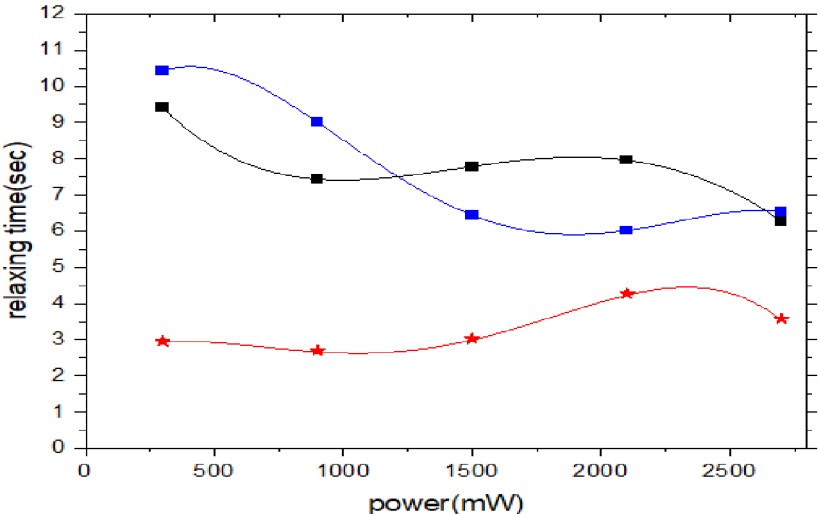

**Figure 7.** Relaxation time constant as a function of the power for the untreated (red), 3 h treated (blue), and 6 h treated (black).

Figure 7 illustrates how changing the time constant with the trap's power predicted interesting behavior for the relaxation rate of the A549 cells. Despite an increase in power, the samples showed that the time constant first decreased before reaching a minimum value and then increased as the power increased. From the three samples, we found that the lung cell relaxation increased with the increasing trap power up to a certain limit. The time constant appeared to increase when the power increased beyond these limits, which means the cell relaxed gradually with the increased power.

According to the three graphs in Figure 7, the time constant reached a minimum of 2.8 s at approximately 900 mW for the lung cells in the untreated sample, a minimum of 6.01 s at approximately 2100 mW for the 3 h treated sample, and a minimum of 6.02 s at approximately 2700 mW for the 6 h treated sample. Overall, although the relaxation time fluctuated with power. Figure 7 shows that the treatment affected how the cell responded to intensity. Overall, the treatment caused the relaxation time to decrease over the power range.

There was no statistically significant difference in the cell relaxing times based on the power ($F_{(4, 10)} = 0.259$, $p = 0.897$). However, there was a statistically significant difference in the cell relaxing times based on the treatment length ($F_{(2, 12)} = 18.224$, $p < 0.001$). Other visible effects (Figure 8) when there was treatment included, on average, the lower the power the higher the cell relaxation; however, this was true more for the 3 h treatment than for the 6 h treatment. This shows an interesting curvilinear-like variable relationship that warrants more investigation.

The relaxation rate (i.e., decrease in the time constant) could be explained by the inevitable heating effect due to the trapping laser's increased power [31]. The wavelength of the infrared laser (1064 nm) could penetrate the water molecules more strongly at relatively high powers, which could increase thermal vibrations within the lung cells, possibly leading to an increase in lung size [7,8,16]. The heating process, however, would compromise the integrity of the cytoskeleton of the lung cells, which is largely responsible for lung cells' viscoelastic properties. The untreated sample appeared more susceptible to this effect than a treated sample. A recent study showed that treating normal cells with suffruticosol B had significantly less toxicity [32].

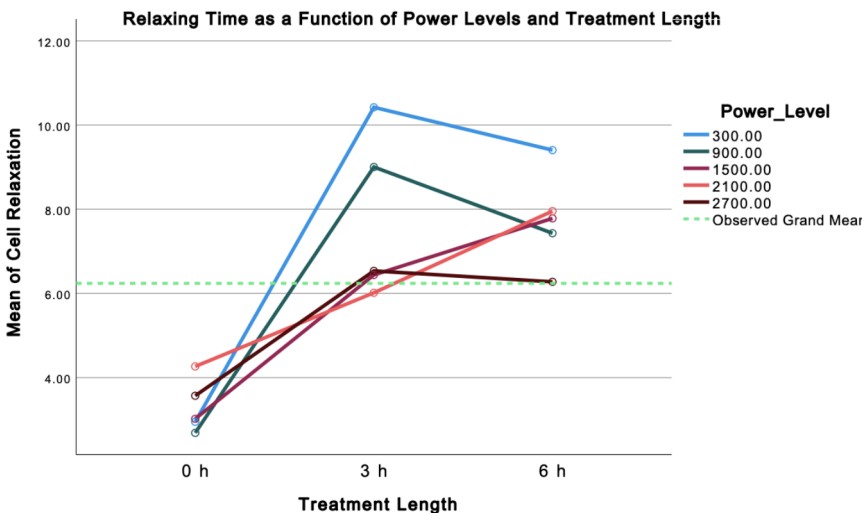

**Figure 8.** Relaxation time constant as a function of the treatment length at various powers.

As the cells became charged and were ejected from the trap at extreme powers, we observed the cell membranes' rupture. Figure 7 shows that the lung cells from the treated samples had a higher time constant at extremely low power. The therapeutic ratio of cells treated with suffruticosol B will be improved.

Since the A549 lung cells were from a single patient, using statistical methods to track the systematic variability in the individual results and synthesized single-subject designs based on experimental replications present a challenge. Thus, the next step will be to obtain these from multiple patients, which would improve this study's experimental technique to acquire more accurate and better predictive results. Our experimental technique utilized a single trap to deform a single cell. As a result, we were limited in the number of cells we could study. With more cells studied, it becomes easier to predict the mechanical properties of a specific sample. It is possible to overcome this limitation by using an acoustic–optical deflector to create multiple traps [33–36]. Such a device allows for trapping multiple cells simultaneously, and we can increase the number of cells studied per sample.

## 5. Conclusions

In this study, suffruticosol B extracted from *Paeonia suffruticosa*, commonly used for antitumor treatment, was applied to cultured human lung carcinoma cells (A549). An intensity gradient laser trap was used to analyze the similarities and differences between untreated, 3 h treated and 6 h treated cells. The cells inside and outside of the trap were measured by their relative sizes. The A549 cell responded interestingly to the treatment in that the untreated sample had a very rapid relaxation time compared to the treated sample. We observed that treatment caused the elasticity of the cell to increase. Our results indicate that both the treatment length and power strength had a significant impact on the cell size, but the effect was greater when the cells were free. However, the effect of treatment on relaxation was more significant than power. Thus, the duration of the treatment had significant effects on its biophysical properties. We used LT to show that the antitumor compound suffruticosol B increased the radiosensitivity of tumor cells.

**Author Contributions:** All authors had full access to the data in the study and take responsibility for the integrity of the data and the accuracy of the data analysis. Conceptualization, D.B.E. and M.S.G.; Methodology, D.B.E. and M.S.G.; Investigation, J.M.R., A.P., J.C., R.M.S., D.L.D. and C.A.B.; Formal Analysis, H.T.C. and M.S.G.; Resources, Y.G., J.M.R., A.P., and D.B.E.; Writing—Original Draft, M.S.G., D.B.E. and H.T.C.; Writing—Review and Editing, H.T.C.; Visualization, D.B.E., M.S.G. and H.T.C.; Supervision, D.B.E. and H.T.C.; Funding Acquisition, H.T.C. All authors have read and agreed to the published version of the manuscript.

**Funding:** Minority Science and Engineering Improvement Program (MSEIP), grant number: P120A200007.

**Data Availability Statement:** All data are available in the main text.

**Conflicts of Interest:** The authors declare no conflict of interest.

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
