# Peer review of "The Effectiveness of Suffruticosol B in Treating Lung Cancer by the Laser Trapping Technique"

_biophysica, doi:10.3390/biophysica3010008_

Round 1
Reviewer 1 Report
The Effectiveness of Suffruticosol B in Treating Lung Cancer by Laser Trapping Technique
Abstract section:
Authors must cite numerical data to support their abstract.
Introduction section.
Eliminate number 3 (subindex)
See line 33
“these3 [3]”.
See line 38
Replace the form of cites [4-8 3-7] by [3-8].
See lines 69 and 70.
Replace Paeonia suffruticosa by Paeonia suffruticosa (in italic form)
Methods section.
The authors must mention where the Suffruticosol B was obtained from, if it was of natural or synthetic origin and write the corresponding reference.
See lines 143 and 144.
“The cells’ Images from the three samples are presented in Fig. 1.”
The cell´s images are presented in Figure 2, but not in figure 1. Please correct this.
Discussion section.
See line 278.
Replace Sufffruticosol by Suffruticosol B.
See line 301 “The program Origin 6.2.” Please write the appropriate reference for the software used.
See line 307. “ in Fig. 8”. Figure 8 is missing please correct this.
Author Response
We would like to thank reviewer the time invested in reviewing this paper and the suggestions that was made.
Reviewer 1
Abstract section:
Authors must cite numerical data to support their abstract.
Response
We have rewrite the abstract to accommodate the reviewer suggestion.
Reviewer 1
Eliminate number 3 (subindex)
See line 33. “these[3]”.
Response
This was fixed. See line 47
Reviewer 1
See line 38.
Replace the form of cites [4-8 3-7] by [3-8].
Response
This was fixed. See line 52
Reviewer 1
See lines 69 and 70.
Replace Paeonia suffruticosa by Paeonia suffruticosa (in italicform).
Response
This was fixed. See lines 89-91
Reviewer 1
Methods section.
The authors must mention where the Suffruticosol B was obtained from, if it was of natural or synthetic origin and write the corresponding reference.
Response
This was fixed. We added lines 113-121 to address this concern.
Reviewer 1
See lines 143 and 144.
“The cells’ Images from the three samples are presented in Fig. 1.”
The cell´s images are presented in Figure 2, but not in figure 1.Please correct this.
Response
This was fixed. See line 257
Reviewer 1
Discussion section.
See line 278.
Replace Sufffruticosol by Suffruticosol B.
Response
This was fixed. See line 397
Reviewer 1
See line 301 “The program Origin 6.2.” Please write the appropriate reference for the software used.
Response
This was fixed. See line 437
REVIEWER 1
See line 307. “ in Fig. 8”. Figure 8 is missing please correct this.
Response
Fixed. This should have been Fig. 7. See line 451
Reviewer 2 Report
In the present paper, the suthors studied the the simultaneous effect of the laser trapping and the Suffruticosol B on the cancel cells A549. Authors varied the time of the treatment and studied the size effect isung the laser trapping. Despite a rather poor set of methods was used, the work is rather unusual and may be of interest for the readership of Biophysica.
The reviewer would suggests to extend the number of the cell lines to make the paper more rigid; otherwise it looks more like a communication.
Otherwise, please change the order of pictures in Figures: a) should be on the top and c) - in the bottom.
Besides, the conclusion must conclude the work. The statement "This cell line responded interestingly to 354 the treatment" should be clarified, and short, but clear explanation of "interestingly" must be given.
Author Response
We would like to thank the reviewer for the time taken to review this manuscript. We have done our due diligence to address the reviewers' concern.
Reviewer 2
The reviewer would suggests to extend the number of the cell lines to make the paper more rigid; otherwise it looks more like a communication.
Response
We agree that more cell lines would make the paper more rigid. We believe that data collected is still interesting and make for a good publication. Further, number of publish articles have look at physical properties on single cell.
Reviewer 2
Otherwise, please change the order of pictures in Figures: a) should be on the top and c) - in the bottom.
Response
This was fixed
Reviewer 2
Besides, the conclusion must conclude the work. The statement "This cell line responded interestingly to 354 the treatment" should be clarified, and short, but clear explanation of "interestingly" must be given.
Response
I believe we address this concern please see lines 467-495, and 531-535
Round 2
Reviewer 2 Report
The paper can be accepted now.